# ONLINE 3D INSTANCE SEGMENTATION AT TASK-ORIENTED GRANULARITY WITH UNPOSED MONOCULAR VIDEO

## ABSTRACT

We present a real-time, task-oriented 3D instance segmentation framework for unposed monocular video, enabling embodied agents to task-adaptively perceive and interact with objects in open-world scenes. Unlike most previous bottom-up segmentation paradigm that segment before recognition, we adopt a task-oriented segmentation approach. Specifically, objects are decoupled within each frame using an open-vocabulary detector combined with a prompt-based 2D segmentation model, while the 3D underlying geometry of the scene is simultaneously being reconstructed using a modern dense SLAM system. Guided by the SLAM-derived pose graph, we selectively associate multi-view masks and reuse the dense correspondences provided by the SLAM system, incrementally converting them into geometric association scores with minimal additional computation. By incorporating semantic similarity and mutual exclusivity metrics, we design a priority-ordered mask clustering algorithm for efficient online multi-view mask matching and merging. Evaluations on open-vocabulary 3D instance segmentation benchmarks show that our method effectively mitigates the performance degradation of existing approaches when using dense SLAM reconstructions instead of depth-sensor point clouds. On the Replica dataset, using only unposed images, it even achieves results comparable to methods leveraging ground-truth depth and poses. Codes will be released upon acceptance of the paper.

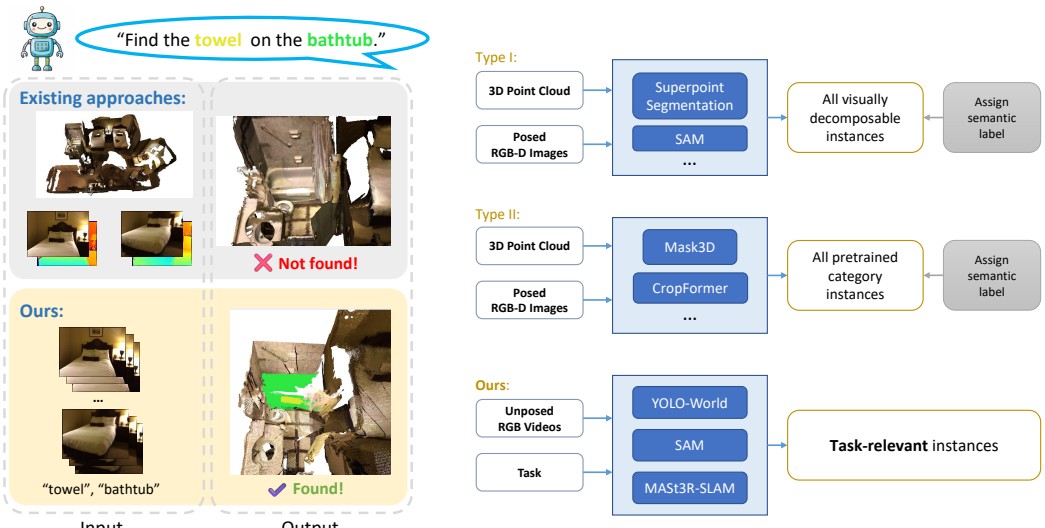

Figure 1: We propose a task-oriented 3D instance segmentation method with online unposed images, without requiring offline-collected point clouds or additional sensor data. Unlike existing bottom-up approaches that segment before recognition, our method determines segmentation granularity based on the task. For example, in the task "Find the towel on the bathtub," conventional methods may falsely merge the towel and bathtub, while ours correctly separates the towel as an individual instance.

# 1 INTRODUCTION

3D instance segmentation is a fundamental task in embodied intelligence, which typically aims to decouple and semantically recognize object instances in a scene using offline-collected sensor data from RGB-D cameras or LiDAR. However, the assumption of having complete offline-collected sensor data rarely holds in embodied intelligence applications, where agents must incrementally perceive, reconstruct, and parse their surroundings under real-time constraints without access to full observations in advance. Moreover, the reliance on depth sensors inherently limits the applicability of these approaches. By contrast, monocular cameras are more cost-effective, easier to deploy, and provide richer semantic information, making them particularly advantageous for a broader range of embodied intelligence applications. Furthermore, unlike traditional settings where instance segmentation is restricted to categorization within predefined taxonomies (e.g., ScanNet Dai et al. (2017)), embodied intelligence requires open-world generalization, where an agent must flexibly adapt its perception to novel tasks and semantic concepts encountered during interaction.

Recently, with the advancement of deep learning, 2D image segmentation and open-vocabulary recognition models such as SAM Kirillov et al. (2023); Ravi et al. (2024), CLIP Radford et al. (2021), and YOLO-World Cheng et al. (2024) have been proposed and widely applied in open-vocabulary 3D instance segmentation tasks. As shown in Fig. 1, existing approaches generally fall into two categories: (1) the first Yang et al. (2023); Yin et al. (2024) leverages models like SAM Kirillov et al. (2023) or superpoint segmentation Felzenszwalb & Huttenlocher (2004) to segment all visually decomposable entities in the scene, but often suffers from fragmented and redundant segmentation, and relies heavily on time-consuming post-processing; (2) the second Tang et al. (2025); Takmaz et al. (2023) depends on pre-trained 2D segmentation Qi et al. (2022) or 3D segmentation models Hou et al. (2023) with closed-set categories to determine segmentation granularity, which cannot continuously adapt segmentation granularity to task requirements. We observe that for each specific embodied intelligence task, the set of object categories that need to be recognized is usually **task-dependent** and **clearly defined.** Consequently, we can move away from the traditional bottom-up segmentation paradigm—where segmentation is performed prior to recognition—toward a task-oriented segmentation paradigm.

To reduce reliance on offline-collected sensor data and pre-known poses, integrating modern monocular SLAM systems can provide real-time 3D scene reconstruction as a basis for instance segmentation. The latest dense SLAM systems, such as MASt3R-SLAM Murai et al. (2025) and VGGT-SLAM Maggio et al. (2025), are capable of performing real-time online reconstruction from unposed monocular video. However, when directly replacing sensor depth and ground-truth poses with the output of these feedforward SLAM methods, existing 3D instance segmentation methods perform poorly. Hence, effectively combining modern 3D dense SLAM methods with 3D scene instance segmentation remains a non-trivial problem.

In this paper, we propose a novel online framework for task-driven 3D instance segmentation from unposed monocular video. Our approach builds upon MASt3R-SLAM, a dense SLAM system, to perform online scene reconstruction directly from the live image stream. For each incoming keyframe, we leverage an open-vocabulary object detector together with a prompt-based 2D segmentation model to decouple and recognize task-relevant objects in 2D images. Guided by the pose graph of MASt3R-SLAM, we avoid redundant pairwise computations between all frames by performing mask association only with the most spatio-temporally relevant keyframes. Moreover, we fully reuse the point-level correspondences computed by the SLAM system, which are **long-term valid** and **independent of the optimized state variables**, enabling efficient incremental computation with minimal additional overhead. We introduce semantic cues and mutual-exclusivity constraints with an online priority-ordered clustering strategy for efficient multi-view mask association and decoupling, and maintain them in an append-only mapping table for fully incremental reconstruction and segmentation. Our method does not rely on any 3D dataset for fine-tuning or post-training, and achieves online, zero-shot, task-oriented 3D instance segmentation.

Our contributions are summarized as follows:

- We propose a novel real-time system capable of task-oriented, specified-granularity, online 3D object segmentation from monocular video without requiring pose information.

- We design a task-relevant 2D instance mask decoupling strategy and reuse associations and correspondence information from the SLAM system, together with our proposed priority-ordered online mask clustering algorithm that effectively merges instance masks across different views.

- We evaluate our method on open-vocabulary 3D instance segmentation benchmarks and show that existing methods struggle to adapt to modern dense SLAM systems. Using only unposed monocular video, our approach outperforms all baselines on the ScanNet200 dataset Dai et al. (2017) and remains competitive with methods using ground-truth depth and poses on the Replica dataset Straub et al. (2019).

## 2 RELATED WORKS

### 2.1 FEED-FORWARD SCENE RECONSTRUCTION

The pioneering work DUSt3R Wang et al. (2024) introduced a feed-forward framework that directly predicts dense point clouds and recovers camera parameters from image pairs. MASt3R Cabon et al. (2025) incorporated feature descriptors for cross-image correspondences, and MASt3R-SFM Duisterhof et al. (2025) extended this approach to multi-view global optimization, although with high computational cost. VGGT Wang et al. (2025a) overcame the two-view limitation via alternating frame-wise and global attention, enabling end-to-end reconstruction from an arbitrary number of views. Spann3R Wang & Agapito (2024) and CUT3R Wang et al. (2025b) further advanced online incremental multi-frame reconstruction, the former leveraging memory mechanisms and the latter employing recurrent temporal networks. MASt3R-SLAM Murai et al. (2025) extended MASt3R Cabon et al. (2025) into a real-time dense SLAM system with globally consistent poses and geometry. In contrast, VGGT-SLAM Maggio et al. (2025) addressed the ambiguity of uncalibrated reconstruction by optimizing 15-DoF homographies across submaps on the SL(4) manifold. Collectively, these works illustrate how feed-forward architectures can evolve into dense SLAM systems; however, they primarily focus on geometric reconstruction, neglecting semantic understanding. Our work builds on these advances to achieve task-driven 3D instance segmentation directly from monocular video, eliminating the need for a depth sensor.

### 2.2 VFM FOR OFFLINE 3D INSTANCE SEGMENTATION.

Benefiting from large-scale 2D annotated data, many vision foundation models (VFMs) have rapidly advanced, showing strong performance and generalization in 2D segmentation and recognition. However, high-quality 3D annotated data remains scarce, motivating researchers to leverage 2D VFMs to assist 3D segmentation and bridge the gap between 2D and 3D understanding. With VFM assistance, many methods achieve strong results in open-vocabulary 3D instance segmentation. Current approaches mainly fall into two categories: (1) SAM3D Yang et al. (2023) and Sai3D Yin et al. (2024) use SAM Kirillov et al. (2023) and superpoint segmentation Felzenszwalb & Huttenlocher (2004) to split all entities in 2D images or 3D point clouds, often suffering from over-segmentation and requiring complex post-processing, while leveraging mask-pooled CLIP embeddings to assign semantic labels to objects; (2) OpenMask3D Takmaz et al. (2023) and Open3DIS Nguyen et al. (2024) use 3D pretrained segmentation Hou et al. (2023); Ngo et al. (2023) models to define granularity, assigning semantics via mask-pooled CLIP Radford et al. (2021) embeddings, while OpenIns3D Huang et al. (2024) and OpenYOLO 3D Boudjoghra et al. (2024) leverage open-vocabulary detection results Cheng et al. (2024); Liu et al. (2024) for semantic labeling. However, all of these methods are limited to closed-set granularity. Different from the above bottom-up or closed-set segmentation strategies, we adopt a top-down segmentation approach with specified task-relevant granularity to better meet the diverse requirements of embodied applications. OVIR-3D Lu et al. (2023) follows a similar paradigm, but it fully relies on sensor depth and poses to establish correspondences Zhou et al. (2022). Moreover, all of these methods are based on offline-collected data, making it difficult to meet the requirements of online embodied applications.

### 2.3 ONLINE 3D INSTANCE SEGMENTATION.

With the rise of embodied AI and the growing demand for diverse robotic applications, online 3D instance segmentation has attracted increasing attention. Early methods McCormac et al. (2017);

Narita et al. (2019) processed 2D images independently, projected predictions onto 3D point clouds, and fused results across frames, but the lack of geometric and temporal information made fusion challenging. Fusion-aware 3D-Conv Zhang et al. (2020) and SVCNN Huang et al. (2021) preserved prior frame information and aggregated 3D features for semantic segmentation. INSCONV Liu et al. (2022) extended sparse convolutions for efficient global 3D feature extraction, while MemAda Xu et al. (2024b) employed multimodal memory adapters for online perception. EmbodiedSAM Xu et al. (2024a) lifted SAM-generated 2D masks to precise 3D masks for high-accuracy per-frame fusion. OnlineAnySeg Tang et al. (2025) merged VFM-generated masks based on spatial alignment, with feature similarity as auxiliary guidance. However, these methods require RGB-D streams with known poses. PanoRecon Wu et al. (2024) and ERrecon Zhou et al. (2025) perform reconstruction and segmentation simultaneously from monocular video but still rely on known poses and models with only closed-set recognition capabilities. Most prior methods rely on offline poses or depth. In online, depth-sensor–free settings, poses and geometry are incrementally estimated and continuously updated, making it hard for existing approaches to adjust segmentation online and rendering them sensitive to noise in these optimization variables. Although modern SLAM methods can provide pose and geometry estimates, these optimization variables are continuously updated online. Above methods cannot revise past segmentations accordingly and are sensitive to noise in these variables. To address this, we propose a pose/geometry-agnostic mask association design that enables online task-oriented 3D instance segmentation from unposed monocular video.

## 3 METHOD

We provided an overview of method in Fig. 2, which shows our main components: MASt3R-SLAM Murai et al. (2025) system(Sec. 3.1), task-oriented mask segmentation(Sec. 3.2), masks association criteria(Sec. 3.3) and online mask merging(Sec. 3.4).

### 3.1 PRELIMINARIES

Our method builds upon MASt3R-SLAM. Given a stream of RGB images $\{I^t \in \mathbb{R}^{H \times W \times 3}\}$ as input, MASt3R-SLAM outputs per-pixel 3D pointmaps $\{X^i \in \mathbb{R}^{H \times W \times 3}\}$ along with their confidences $\{C^i \in \mathbb{R}^{H \times W \times 1}\}$ of keyframes $\{\mathcal{K}^i\}$ and camera pose $\{T^t \in Sim(3)\}$of all frames.

A key component of MASt3R-SLAM is pointmap matching, which establishes dense pixel correspondences: $\pi_{ij} : p_i \rightarrow p_j$ as well as the valid mask $V_{ij} \in \mathbb{B}^{H \times W \times 2}$ between frames, where $p_i \in \mathbb{R}^{H \times W}$ represents the pixel coordinates of image $i$ and the valid mask $V_{ij} = False$ denotes the invalidate matches with large distances in 3D space or low predicted confidence scores. By formulating correspondence search as a local non-linear optimization and leveraging GPU parallelization, MASt3R-SLAM achieves highly efficient pointmap matching. Importantly, the matching is completely **independent of pose estimates**, relying solely on MASt3R outputs. This ensures that correspondences remain valid even as back-end optimisation and loop closure continuously refine keyframe poses.

To maintain consistency over long sequences, MASt3R-SLAM constructs an incremental pose graph $\mathcal{E}$. Each incoming frame is compared against the latest keyframe $\mathcal{K}^{i-1}$, and a new keyframe $\mathcal{K}^i$ is added when geometric matches fall below a threshold. Graph edges are formed either sequentially or through loop closure detection.

In summary, MASt3R-SLAM provides (1) dense and reliable point-level correspondences across frames and (2) keyframe mechanism and factor graph that captures spatiotemporal associations between keyframes. These two properties are particularly important for our extension: we leverage the point-level matching to associate instance masks across frames, and we exploit the incremental and sparse nature of the factor graph to selectively and efficiently compute associations of instance masks between different frames.

### 3.2 TASK-ORIENTED MASK REPRESENTATION

For each new added keyframe $\mathcal{K}^i$, we adopt a task-oriented instance mask generation strategy. Specifically, given a task-relevant open-category set $\mathcal{C}$(e.g., set {towel, bathtub} for "Find the towel on the bathtub."), which can be inferred automatically by an LLM Achiam et al. (2023); Touvron

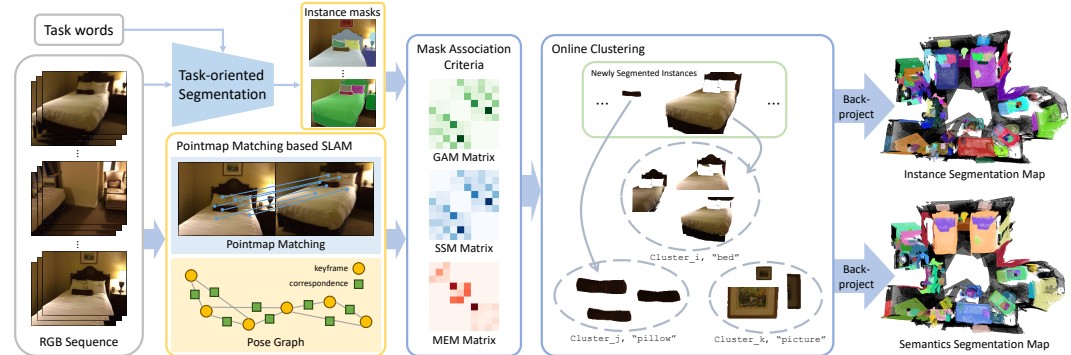

Figure 2: **System diagram of ours**. Upon arrival of a new frame, keyframes are selected and the pose graph is dynamically updated via MASt3R-SLAM, with point correspondences established. Task-oriented instance masks are extracted for newly added keyframes. Guided by the pose graph and inter-frame correspondences, mask association criterias are incrementally computed. Followed by online clustering that establishes cross-frame instance identities, the cross-frame instance masks as well as corresponding semantic labels are back-projected onto the reconstructed point cloud to obtain 3D instance and semantic maps.

et al. (2023); Yang et al. (2025), YOLO-World Cheng et al. (2024) first generates coarse, category-aware 2D proposals. These proposals are then refined by SAM2 Ravi et al. (2024) to yield high-quality instance masks. In accordance with the small-mask retention priority principle(See App.A.1 for details.), we can process the instance masks of each keyframe to be non-overlapping. Therefore, the instance masks of each keyframe can be parameterized as $\mathbb{M}^i \in \mathbb{Z}^{H \times W}$, where $M^i_{id=n}$ refer to the n-th detected instance mask in the current scene, $c^i_n \in \mathcal{C}$ is its associated category label, and $M^i_{id=-1}$ represents the background region.

Since the pointmap $X^i$ and instance mask representation $M^i$ of keyframes are pixel-wise aligned, we can equivalently formulate the 3D instance segmentation problem of pointmap as an instance masks association problem across multiple keyframes. Once multi-view association finished, the instance IDs of pointmap across views are updated via a single-level hash mapping rule.

### 3.3 MASKS ASSOCIATION CRITERIA

To efficiently establish correspondences among instance masks across frames, we leverage the incremental and sparse nature of the pose graph $\mathcal{E}$ to guide online masks association computation, avoiding redundant and irrelevant frame-to-frame association. Specifically, whenever a new edge $e_{ij} = \left(\mathcal{K}^i, \mathcal{K}^j\right)$ is added to the pose graph, we compute pairwise mask associations between the mask sets $\left\{M^i_{id=n}\right\}$ and $\left\{M^j_{id=m}\right\}$ of the two endpoint keyframes. For each mask pair $(M^i_{id=n}, M^j_{id=m})$, we compute and record the geometric association metric and semantic similarity metric. In addition, for the newly added keyframe $\mathcal{K}^i$, we impose mutual exclusivity constraints among masks within its set $\mathbb{M}^i$.

**Geometric association metric.** For each new edge $e_{ij} = \left(\mathcal{K}^i, \mathcal{K}^j\right)$, we have establishes bidirectional dense pixel correspondences between them: $\pi_{ij} : p_i \rightarrow p_j$ and $\pi_{ji} : p_j \rightarrow p_i$. For each mask pair $(M^i_{id=n}, M^j_{id=m})$ from keyframe $\mathcal{K}^i$ and $\mathcal{K}^j$, we use correspondence $\pi_{ij}$ to project the mask $M^i_{id=n}$ to the image coordinate frame of keyframe $\mathcal{K}^j$, denoted as $\pi_{ij}(M^i_{id=n})$. We can then compute the overlap between $\pi_{ij}(M^i_{id=n})$ and $M^j_{id=m}$, yielding the intersection $\pi_{ij}(M^i_{id=n}) \cap M^j_{id=m}$. With this, the valid overlap ratio of $M^i_{id=n}$ to $M^j_{id=m}$ is defined as follows:

$$or_{(n,m)} = \frac{\left|(\pi_{ij}(M^i_{id=n}) \cap M^j_{id=m}, V_{ij})\right|}{\left|(M^i_{id=n}, V_{ij})\right|} \tag{1}$$

Where $|(M, V)|$ counts the number of pixel correspondences whose source pixels lie inside mask M and are marked valid by $V$ (i.e., $V_{ij} = $ True). Object observations during online mapping are often incomplete. If $M_{id=n}^i$ and $M_{id=m}^j$ correspond to the same object but $\left|M_{id=n}^i\right| \gg \left|M_{id=m}^j\right|$, the overlap ratio $or(n, m)$ may be underestimated due to inevitable matching noise. Therefore, we also compute $or(m, n)$, which represents the valid overlap ratio of $M_{id=m}^j$ to $M_{id=n}^i$. The final geometric association metric between these two masks is defined as the maximum of these two overlap ratios:

$$GAM_{(n,m)} = GAM_{(m,n)} = max(or_{(n,m)}, or_{(m,n)}) \in [0, 1] \tag{2}$$

**Semantic similarity metric.** Similar to the geometric association metric, , the semantic similarity metric is computed only between mask sets of two keyframes connected by a pose graph edge. We pre-compute text features for each category in the task-relevant open-category set $\mathcal{C}$ using CLIP only once, denoted as: $\mathcal{F} = \left\{f_c \,|\, c \in \mathcal{C}, \ f_c \in \mathbb{R}^d\right\}$, where $f_c$ is the text embedding of category $c$. Each instance $\text{mask}(M_n^i, c_n^i)$ directly takes its semantic feature from the corresponding category embedding:$s_n^i = f_{c_n^i}$. Compared with methods that crop instances and extract image features via CLIP, this approach significantly reduces computation. For a mask pair$(M_{id=n}^i, M_{id=m}^j)$, the semantic similarity metric is defined as the cosine similarity of their semantic features:

$$SSM_{(n,m)} = SSM_{(m,n)} = \frac{\left\langle s_n^i, s_m^j \right\rangle}{\left\|s_n^i\right\|\left\|s_m^j\right\|} \in [-1, 1] \tag{3}$$

where $\langle ., . \rangle$ denotes the inner product and $\|.\|$ the Euclidean norm.

**Mutual exclusivity metric.** While geometric and semantic cues promote mask merging, they may also incorrectly associate mismatched masks—for instance, due to under-segmentation where a bounding box intended for one object may inherently include parts of others, or due to noisy correspondences. To prevent incorrect associations, we propose a mutual exclusivity score for intra-frame masks, based on the premise that different instance masks within a keyframe should not be merged. When a new keyframe $\mathcal{K}^i$ is added, before the explicit non-overlapping constraint is enforced (see Sec. 3.2), some instance masks within the keyframe $\mathcal{K}^i$ may still partially overlap. For mask pairs with negligible overlap, we define the mutual exclusivity score as following:

$$MEM_{(n,m)} = MEM_{(m,n)} = \begin{cases} 1 & iou\left(\hat{M}_n^i, \hat{M}_m^i\right) < \epsilon \\ 0 & otherwise \end{cases} \tag{4}$$

where $\epsilon$ is a threshold, $\hat{M}$ denotes the instance masks before applying the explicit non-overlapping constraint.

As shown in Fig. 2, these pairwise mask metrics are stored in a dense array, where the entry at row $i$ and column $j$ encodes the association score between the $i$-th and $j$-th masks. Since random access in a dense array has $O(1)$ time complexity, both lookups and updates can be performed in constant time, enabling efficient access during online processing.

### 3.4 MASK MERGING BASED ON THE PRIORITY-ORDERED PRINCIPLE

**Masks merging based on association criterias.** We model multi-view mask matching as an online clustering problem over masks. A cluster $\gamma$ is a set of masks that represent the same object. Whenever a new set of edges $\{e_{ij}\}$ is added to the pose graph and the mask association criteria is computed, the new mask edges $\{(M_{id=n}^i, M_{id=m}^j, GAM_{(n,m)}, SSM_{(n,m)})\}$ will be added to the mask edge set $\mathbf{E}$. Then we need to update assignment state of instance masks based on all available association information. The detailed procedure is provided in Alg. 1. Briefly, for all newly added mask pairs, we first select plausible candidate matches by applying thresholds to the geometric and semantic similarity metrics. These candidate pairs are then processed in descending order of their geometric matching scores, giving priority to pairs with higher matching confidence. During mask assignment and cluster merging, we refer to the mutual exclusivity matrix to ensure that no mutually exclusive mask pairs coexist within the same cluster.

**Clusters merging based on IoU criteria.** Since the computation of association metrics rely on the pose graph, keyframes that are temporally distant but spatially close may lack connecting edges, making it hard to associate masks of the same instance. To address this, we use the first-stage clustering results ($num_{clusters} \gg num_{masks}$) to compute 3D bounding boxes for each cluster and their pairwise IoUs. Following a principle similar to that in Algorithm 1, clusters are then merged in descending IoU order, while consulting the mutual exclusivity matrix to ensure no conflicting masks are merged.

---

**Algorithm 1** Online Mask merging based on the priority-ordered principle.

---

**Input:** $\mathbf{E} = \{(M_{id=n}^i, M_{id=m}^j, GAM_{(n,m)}, SSM_{(n,m)})\}, \mathbf{M} = \{MEM_{(n,m)}\}$, clusters $\Gamma = \{\gamma_k\}$
**Output:** processed mask edges $\mathbf{E}^*$, clusters $\Gamma^*$
1: $\Gamma^* \leftarrow \Gamma, \mathbf{E}^* \leftarrow \texttt{filter\_mask\_edges}(\mathbf{E}, \mathbf{S})$
2: $\mathbf{E}^* \leftarrow \texttt{sort}(\mathbf{E}^*, \texttt{keyword=}\{GAM_{(n,m)}\}, \texttt{descend})$
3: **for** $(M_{id=n}^i, M_{id=m}^j, GAM_{(n,m)}, SSM_{(n,m)}) \in \mathbf{E}^*$ **do**
4:     $\gamma_n \leftarrow \texttt{get\_cluster}(M_{id=n}^i), \gamma_m \leftarrow \texttt{get\_cluster}(M_{id=m}^j)$
5:     **if** $\gamma_n = \emptyset$ **and** $\gamma_m = \emptyset$ **then**
6:        $\gamma_{new} \leftarrow \{M_{id=n}^i, M_{id=m}^j\}$              ▷ create a new cluster with the masks
7:        $\Gamma^* \leftarrow \{\gamma_{new}\}$
8:     **else if** $\gamma_n \neq \emptyset$ **and** $\gamma_m \neq \emptyset$ **then**
9:        **if** $\gamma_n \neq \gamma_m$ **then**
10:           **if** $\texttt{is\_violate}(\gamma_n, \gamma_m, \mathbf{M})$ **then**
11:              $\mathbf{E}^* \leftarrow \mathbf{E}^* \setminus \{(M_{id=n}^i, M_{id=m}^j, GAM_{(n,m)}, SSM_{(n,m)})\}$     ▷ discard the edge
12:           **else**
13:              $\Gamma^* \leftarrow \Gamma^* \setminus \{\gamma_n, \gamma_m\}$                 ▷ delete the two clusters
14:              $\Gamma^* \leftarrow \Gamma^* \cup \{\gamma_n \cup \gamma_m\}$               ▷ merge the two clusters
15:           **end if**
16:        **end if**
17:     **else**
18:        # assuming $\gamma_n \neq \emptyset, \gamma_m = \emptyset$, vice versa.
19:        **if** $\texttt{is\_violate}(\gamma_n, M_{id=m}^j)$ **then**
20:           $\mathbf{E}^* \leftarrow \mathbf{E}^* \setminus \{(M_{id=n}^i, M_{id=m}^j, GAM_{(n,m)}, SSM_{(n,m)})\}$     ▷ discard the edge
21:        **else**
22:           $\gamma_n^* \leftarrow \gamma_n \cup \{M_{id=m}^j\}$       ▷ add the unassigned mask to the cluster
23:           $\Gamma^* \leftarrow \Gamma^* \setminus \{\gamma_n\}$              ▷ remove the older cluster
24:           $\Gamma^* \leftarrow \Gamma^* \cup \{\gamma_n^*\}$               ▷ add the updated cluster
25:        **end if**
26:     **end if**
27: **end for**

---

## 3.5 Implementation Details

During the task-oriented mask detection stage, we discard segmentation results where the bounding box of YOLO-World and the corresponding instance mask from SAM2 have low overlap. All experimental scenes share the same set of hyperparameters: the threshold $\epsilon$ for determining mutual exclusivity is set to 0.2, while the thresholds for the Geometric association metric($\tau_{GAM}$), Semantic similarity metric($\tau_{SSM}$), and inter-cluster IoU($\tau_{IoU}$) filtering are 0.25, 0.85, and 0.1, respectively.

## 4 Experiments

In this section, we conduct extensive experiments to evaluate our method against state-of-the-art approaches on publicly available 3D instance segmentation datasets. We begin by describing the experimental setup (Sec. 4.1), followed by quantitative results and qualitative analyses (Sec. 4.2). And we present an ablation study (Sec. 4.3) to demonstrate the effectiveness of our key designs. Finally, we provide a detailed report on the runtime analyzes.

## 4.1 EXPERIMENTAL SETUP

Table 1: **Open-vocabulary and class-agnostic 3D instance segmentation quantitative result on the ScanNet200 validation set.** We also attempted to replace the inputs of some baseline methods with point clouds and camera poses estimated by MASt3R-SLAM. Under this setting, our method significantly outperforms these baselines.

| Method | Online | Zero-shot | Segment Granularity | Open-Vocabulary | Pose&Depth | AP50↑ | AP25↑ | FPS↑ |
|---|---|---|---|---|---|---|---|---|
| SAM3D | × | ✓ | SAM | CLIP | GT | 14.2 | 21.3 | - |
| SAM3D | ✓ | ✓ | SAM | CLIP | GT | 14.7 | 19.0 | 8 |
| OVIR-3D | × | ✓ | Detic | Detic | GT | 24.9 | 32.3 | - |
| Open3DIS | × | × | ISBNet | CLIP | GT | 29.4 | 32.8 | - |
| OpenIns3D | × | × | Mask3D | Grounding Dino | GT | 10.3 | 14.4 | - |
| OpenMask3D | × | × | Mask3D | CLIP | GT | 19.9 | 23.1 | - |
| Open-YOLO 3D | × | × | Mask3D | YOLO-World | GT | **31.7** | **36.2** | - |
| EmbodiedSAM | ✓ | × | ScanNet200 | CLIP | GT | 19.2 | 23.9 | 10 |
| Open-YOLO 3D | × | × | Mask3D | YOLO-World | MASt3R-SLAM | 0.8 | 2.0 | - |
| OnlineAnySeg | ✓ | ✓ | CropFormer | CLIP | MASt3R-SLAM | 0.1 | 0.5 | **15** |
| Ours | ✓ | ✓ | YOLO-World + SAM | YOLO-World | MASt3R-SLAM | **6.5** | **18.7** | 7 |
| OVIR-3D | × | ✓ | Detic | Class-Agnostic | GT | 27.5 | 38.8 | - |
| SAM3D | ✓ | ✓ | SAM | Class-Agnostic | GT | 24.8 | 49.6 | 8 |
| EmbodiedSAM | ✓ | × | ScanNet200 | Class-Agnostic | GT | **65.4** | **80.9** | 10 |
| OnlineAnySeg | ✓ | ✓ | CropFormer | Class-Agnostic | GT | 36.1 | 53.5 | **15** |
| OnlineAnySeg | ✓ | ✓ | CropFormer | Class-Agnostic | MASt3R-SLAM | 4.7 | 19.5 | **15** |
| Ours | ✓ | ✓ | YOLO-World + SAM | Class-Agnostic | MASt3R-SLAM | 16.0 | 44.1 | 7 |

Table 2: Open-vocabulary 3D instance segmentation quantitative result on the Replica dataset.

| Method | Online | Zero-shot | Segment Granularity | Open-Vocabulary | Pose&Depth | AP50↑ | AP25↑ |
|---|---|---|---|---|---|---|---|
| OVIR-3D | × | ✓ | Detic | Detic | GT | 20.5 | 27.5 |
| Open3DIS | × | × | ISBNet | CLIP | GT | 24.5 | 28.2 |
| OpenMask3D | × | × | Mask3D | CLIP | GT | 18.4 | 24.2 |
| OpenScene | × | × | Mask3D | LSeg | GT | 15.6 | 17.3 |
| Open-YOLO 3D | × | × | Mask3D | YOLO-World | GT | **28.6** | 34.8 |
| Ours | ✓ | ✓ | YOLO-World + SAM | YOLO-World | MASt3R-SLAM | 23.4 | **37.3** |

We conduct our experiments using the ScanNet200 and Replica datasets. Our analysis on ScanNet200 is based on its validation set, comprising 312 scenes. For the 3D instance segmentation task, we utilize the 198 predefined categories from the ScanNet200 annotations. Additionally, we conduct experiments on the Replica dataset, which contains 48 categories. Since our method takes only monocular videos without poses as input, we use evo Grupp (2017) to align the estimated trajectory with the ground truth via a Sim(3) transformation, which is then applied to the reconstructed point cloud to align it with the ground-truth mesh. We then transfer semantic and instance labels of the reconstructed point cloud to the ground-truth mesh through nearest-neighbor vertex lookup.

Table 3: Ablation study with different design in mask association and merging on the Replica dataset.

| Method | AP50↑ | AP25↑ |
|---|---|---|
| w/o GAM | 17.3 | 31.5 |
| w/o SSM | 21.7 | 33.1 |
| w/o MEM | 8.2 | 12.0 |
| w/o IoU | 13.8 | 31.4 |
| w/o priority-ordered | 20.3 | 33.8 |
| Our final system | **23.4** | **37.3** |

For evaluation metrics, we follow the ScanNet methodology and report average precision (AP) at two mask overlap thresholds: 50% and 25%. In the open-vocabulary setting, AP evaluates both instance segmentation quality and correct category assignment, whereas in the class-agnostic setting, it evaluates instance segmentation only.

## 4.2 RESULTS ANALYSIS

We conduct comparisons against other methods on the ScanNet200 Dai et al. (2017) and Replica datasets Straub et al. (2019). The evaluation results on ScanNet200 are presented in Tab. 1, where

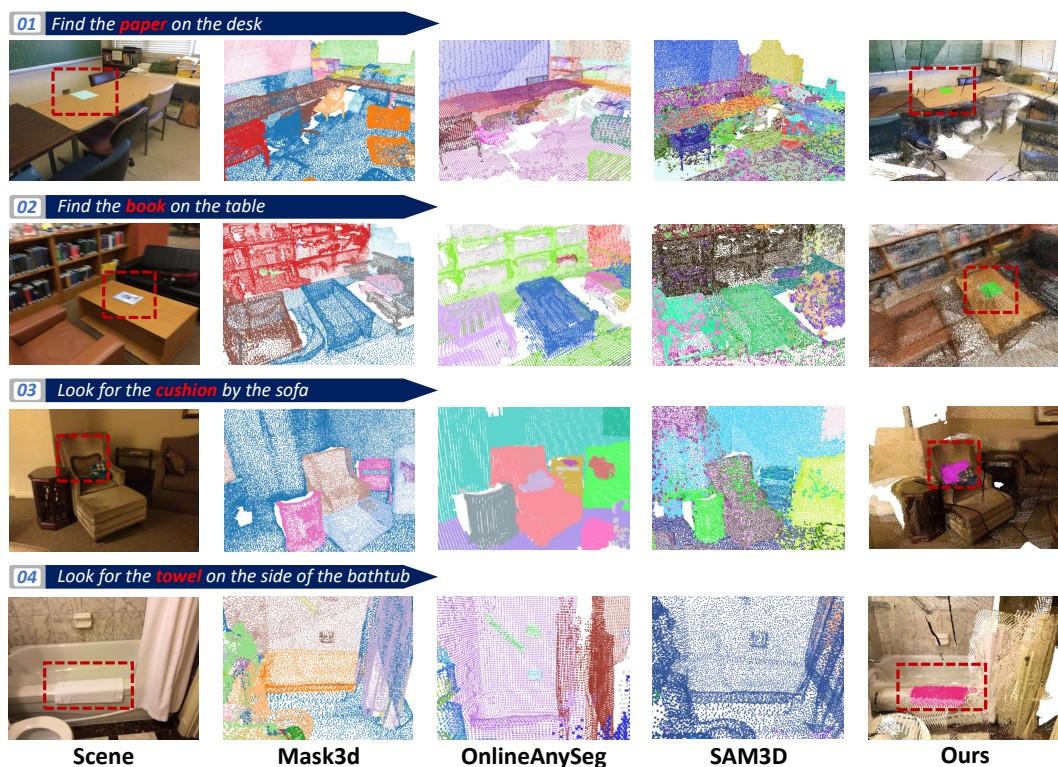

Figure 3: **Qualitive results on the ScanNet dataset.** We show class-agnostic masks from three other bottom-up methods. Unlike them, which often merge small objects into larger neighbors, our approach reliably achieves task-oriented 3D object disentanglement from monocular video. we indicate each method's priors for segmentation granularity and open-vocabulary recognition, its ability to adapt to online segmentation, and its reliance on ground-truth poses and sensor depth.

Under the open-vocabulary setting, our method still lags behind approaches using ground-truth poses and depth, mainly because MASt3R-SLAM reconstructions deviate from ground-truth point clouds, producing mismatched regions and artifacts. This leads to degraded performance on small objects, especially for long-tail categories. At the same time, we observe that when using MASt3R-SLAM reconstructed point clouds and estimated poses as inputs, Open-YOLO 3D performs poorly due to the limited generalization ability of its 3D segmentation network, while OnlineAnySeg heavily relies on high-quality point clouds for mask association and is sensitive to artifacts. In contrast, our mask association strategy demonstrates stronger robustness and reliability, achieving results that substantially surpass these methods. Moreover, under the ScanNet200 Class-Agnostic setting, our method remains competitive without relying on ground-truth poses or depth.

Furthermore, we conduct open-vocabulary instance segmentation experiments on the Replica dataset. As shown in Tab. 2, even without ground-truth camera poses or depth information, our method ranks first on the AP25 metric and achieves highly competitive results on the AP50 metric.

In addition, Fig. 3 presents qualitative results on the ScanNet200 dataset, including class-agnostic instance segmentation results produced by three bottom-up segmentation paradigms: Mask3D Hou et al. (2023), OnlineAnySeg Tang et al. (2025), and SAM3D Yang et al. (2023). Due to ambiguity in segmentation granularity, bottom-up paradigms tend to merge small objects with adjacent larger ones, failing to satisfy task-specific decoupling requirements. In contrast, our task-oriented instance segmentation paradigm adapts effectively to diverse task demands. In instruction-driven applications, the correctness of task-relevant granularity matters more than the AP metric on common benchmark.

## 4.3 ABLATION STUDY

Since mask association and merging are the key components of our method, we conducted experiments to evaluate the effectiveness of different designs. As shown in Tab. 3, we first validate three

mask association criteria: the Geometric Association Metric (GAM), Semantic Similarity Metric (SSM), and Mutual Exclusivity Metric (MEM). Removing any of these criteria results in a noticeable performance drop, with the most severe degradation occurring when MEM is omitted. This is driven by two independent, module-intrinsic factors: (1)inevitable point-matching noise in the SLAM system, and (2) a few under-segmented masks. Without MEM, either factor can cause multiple distinct instances to be erroneously merged into a single instance(see Fig. 4 in App. A.7). Furthermore, removing the IoU-based mask cluster merging impairs the association of masks belonging to the same instance across different time spans(see Fig. 5 in App. A.7). Finally, we evaluate the impact of discarding the priority-ordered algorithm principle during mask merging, i.e., processing mask pairs in random order rather than by descending confidence. The results show a significant drop in segmentation performance, primarily because prioritizing high-confidence mask pairs, in combination with the exclusivity metric, helps mitigate noisy matches and segmentation errors.

### 4.4 RUNTIME ANALYSES

We conduct runtime evaluation on a desktop computer equipped with an Intel i9-12900KS CPU and an NVIDIA RTX 3090 GPU. Our algorithm achieves 10.87 FPS on the scene $office0$ of the Replica dataset and 7.32 FPS on the $scene0011\_00$ of the ScanNet dataset. For reference, Mast3r-SLAM runs at 11.23 FPS on Replica $office0$ and 7.58 FPS on ScanNet $scene0011\_00$. Since our mask association module largely reuses the corresponding computation results from Mast3r-SLAM, both the association step and the system integration incur almost no extra computational overhead. Taking the scene $office0$ of the Replica dataset as an example, we report the runtime breakdown of all major components, as shown in the Tab. 4. Specifically, the average processing time per keyframe is 30.1 ms for YOLO-World Cheng et al. (2024) and 132.4 ms for SAM2 Ravi et al. (2024), while the geometric association metric requires only 9 ms per keyframe pair, and the mask merging step takes 32 ms.

Table 4: Runtime analysis of different components.

| Component | Runtime(ms) |
| --- | --- |
| YOLO-World | 30.1 |
| SAM2 | 132.4 |
| GAM Cal. | 9.0 |
| Masks merging | 32 |

## 5 LIMITATION AND FUTURE WORK

Our current approach is limited to static 3D scenes, and extending it to dynamic scenes by integrating monocular 4D reconstruction methods would be valuable. In addition, the SLAM module in our framework could better leverage the richer semantic and instance information provided by the recognition module, and we view this as a promising extension. Possible directions include (but are not limited to): semantics-guided dynamic suppression, semantic loop-closure proposal generation, and object-level local reconstruction optimization, and we leave these for future work.

## 6 CONCLUSION

We present a real-time, task-oriented 3D instance segmentation framework for unposed monocular video, enabling embodied agents to task-adaptively perceive and interact with objects in open-world scenes. By combining task-oriented 2D instance segmentation with modern dense SLAM-based 3D reconstruction and an efficient online multi-view mask clustering algorithm, our method supports flexible task-oriented 3D instance segmentation. Evaluations demonstrate that our approach overcomes the challenges faced by existing methods when using point clouds reconstructed by dense SLAM, outperforming baseline methods on the ScanNet200 dataset and achieving performance comparable to methods that rely on ground-truth depth and poses on the Replica dataset. Our framework provides a robust solution for online, task-adaptive 3D perception, empowering embodied agents with flexible and adaptive scene understanding.

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

# A APPENDIX

## A.1 NON-OVERLAPPING WITH SMALL-MASK RETENTION

SAM2 may produce multiple instance masks per keyframe that are partially overlap. Since each pixel should belong to exactly one instance, we compose a non-overlapping mask set $\mathbb{M}^i \in \mathbb{Z}^{H \times W}$ using a small-mask retention rule that preserves fine-grained instances (e.g., the 'towel' will not be overwritten by the 'bathtub').

Concretely, we sort all instance masks of the keyframe in descending area order (large to small). Then we write the sorted instance masks of a keyframe into a non-overlapping mask following this rule: (1) if the two masks share the same category label, we treat the smaller one as an over-segmentation part of the larger one and discard the the smaller mask; (2) if the two masks have different category labels, we keep the smaller mask on the overlapping pixels and truncate the larger mask accordingly.

## A.2 REPRODUCIBILITY STATEMENT

We will provide a demo video on `https://anonymous.4open.science/r/OTO-3DIS-FEF8/` and release the source code upon paper acceptance.

## A.3 HYPERPARAMETER SETTINGS

In our experiments, the thresholds for determining mutual exclusivity ( $\epsilon$), the geometric association metric ($\tau_{GAM}$), the semantic similarity metric ($\tau_{SSM}$), and the inter-cluster IoU($\tau_{IoU}$) filtering are set to 0.2, 0.25, 0.85, and 0.1, respectively. These values were selected based on ablation studies. Tables 5, 6, 7, and 10 report the ablation results for these four hyperparameters, showing that our current settings are optimal and that the system's performance is not sensitive to their exact values. The overall metrics vary only slightly across different settings, indicating the robustness of our method.

## A.4 DISCUSSION ON DIFFERENT PERCEPTION OR SLAM METHOD

In response to the reviewer's comments, we discuss whether the perception module and the SLAM module in our current method can be replaced with alternative methods.

**Replacing the open-vocabulary detector.** We directly replaced YOLO-World-xl with YOLO-UniOW-l (the largest publicly available variants of each family) without modifying any parameters of our system. On the Replica dataset, YOLO-World-xl is intrinsically stronger than YOLO-UniOW-l, so this substitution results in a moderate performance drop. Nevertheless, without any additional adaptation(all hyperparameters are kept unchanged.), the system remains fully functional and produces reasonable outputs, demonstrating that the detector module is readily replaceable.

**Replacing the SLAM backbone with VGGT-SLAM.** Like MASt3R-SLAM, VGGT-SLAM can perform online pose estimation and pointmap reconstruction from unposed monocular video. Concretely, VGGT-SLAM partitions long sequences into overlapping submaps and performs direct submap registration/alignment; in its standard form, it does not expose frame-to-frame dense pixel correspondences, which our association module relies on. Therefore, a drop-in replacement with VGGT-SLAM is not currently supported. In principle, however, any SLAM that provides edge-local dense correspondences (e.g., optical flow, feature matches, or pointmap matches) can serve as our backbone.

## A.5 LLM USAGE

Large language model (LLM) were used as an auxiliary tool during the writing and language refinement of this manuscript. Its role was limited to improving linguistic expression, readability, and overall flow, including tasks such as sentence rephrasing and grammar checking.

It is important to emphasize that the LLM did not participate in the research conception, methodological design, or experimental procedures. All research ideas, technical approaches, and data analyses were independently developed and conducted by the authors. The LLM's contribution was strictly confined to language-level improvements and did not involve any scientific content.

The authors take full responsibility for the entire manuscript, including the portions generated or refined with LLM assistance. We have ensured that all content complies with academic ethical standards and does not involve plagiarism or any form of scientific misconduct.

## A.6 FAILURE CASE ANALYSIS

To better analyze the errors of our system on the ScanNet200 benchmark, we follow the official ScanNet200 split based on the frequency of labeled surface points, dividing the 200 classes into 66, 68, and 66 categories, corresponding to head, common, and tail classes, respectively. We report the metrics for these subsets in Table 10.

Table 5: Ablation study with different thresholds for determining mutual exclusivity ($\epsilon$) on the Replica dataset.

| Method | AP50↑ | AP25↑ |
|---|---|---|
| $\epsilon$ =0.1 | 23.3 | 37.2 |
| $\epsilon$ =0.3 | 23.3 | **37.3** |
| $\epsilon$ =0.4 | 23.2 | 37.0 |
| $\epsilon$ =0.2 | **23.4** | **37.3** |

Table 6: Ablation study with different thresholds for geometric association metric ($\tau_{GAM}$) on the Replica dataset.

| Method | AP50↑ | AP25↑ |
|---|---|---|
| $\tau_{GAM}$ =0.15 | 23.3 | 37.1 |
| $\tau_{GAM}$ =0.2 | **23.4** | 37.2 |
| $\tau_{GAM}$ =0.3 | **23.4** | **37.3** |
| $\tau_{GAM}$ =0.35 | **23.4** | 37.2 |
| $\tau_{GAM}$ =0.25 | **23.4** | **37.3** |

Table 7: Ablation study with different thresholds for semantic similarity metric ($\tau_{SSM}$) on the Replica dataset.

| Method | AP50↑ | AP25↑ |
|---|---|---|
| $\tau_{SSM}$ =0.75 | 23.2 | 35.1 |
| $\tau_{SSM}$ =0.8 | 23.2 | 34.2 |
| $\tau_{SSM}$ =0.9 | **23.4** | 37.1 |
| $\tau_{SSM}$ =0.95 | **23.4** | 37.1 |
| $\tau_{SSM}$ =0.85 | **23.4** | **37.3** |

Table 8: Ablation study with different thresholds for inter-cluster ($\tau_{IoU}$) on the Replica dataset.

| Method | AP50↑ | AP25↑ |
|---|---|---|
| $\tau_{IoU}$ =0.05 | 23.4 | **37.3** |
| $\tau_{IoU}$ =0.15 | **23.5** | 37.2 |
| $\tau_{IoU}$ =0.2 | 20.8 | 34.5 |
| $\tau_{IoU}$ =0.1 | 23.4 | **37.3** |

Table 9: Comparison using different open-vocabulary detectors on the Replica dataset.

| Method | AP50↑ | AP25↑ |
|---|---|---|
| Ours with YOLO-UniOW-l | 17.4 | 27.4 |
| Ours with YOLO-World-xl | **23.4** | **37.3** |

Table 10: Open-vocabulary 3D instance segmentation quantitative result of different category splits On the ScanNet200 validation set.

| Method | AP50↑ | AP25↑ |
|---|---|---|
| Head | 9.3 | 24.5 |
| Common | 5.5 | 17.4 |
| Tail | 4.1 | 13.5 |
| Average | 6.5 | 18.7 |

## A.7 QUALITATIVE ABLATION STUDY

**01** *The **pillows** on the bed*

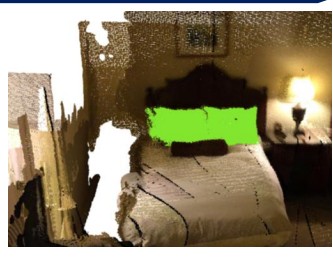 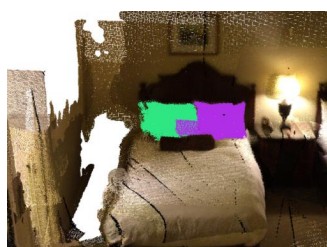

**02** *The **chairs** along the long side of the table*

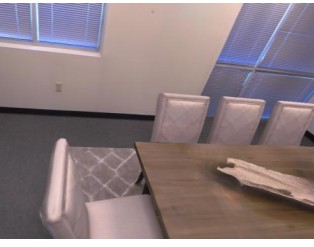 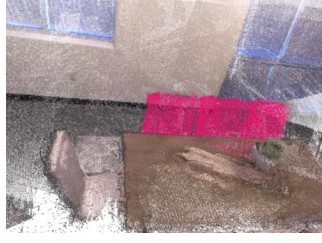 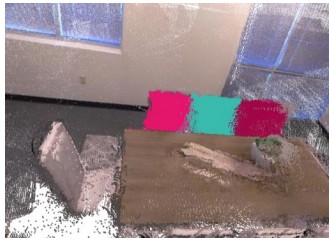

**Scene** **w/o MEM** **with MEM**

Figure 4: **Qualitative ablation study of the MEM metric.** When the MEM mechanism is absent, SLAM matching errors and mask under-segmentation often cause different objects that are adjacent in the scene and share the same semantic label to be merged into a single instance, such as side-by-side pillows or chairs. When the MEM mechanism is enabled, these adjacent objects with the same semantics can be correctly separated and reconstructed as independent instances.

**01** *The **bag** against the wall*

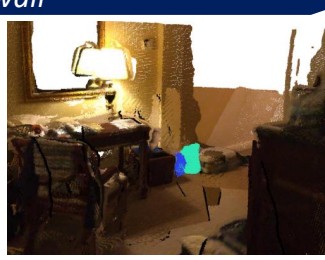 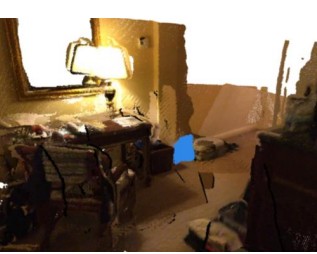

**02** *The **stool** near the door*

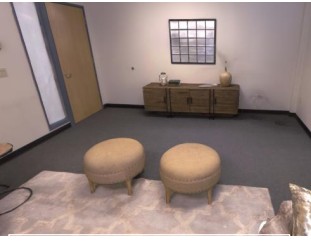 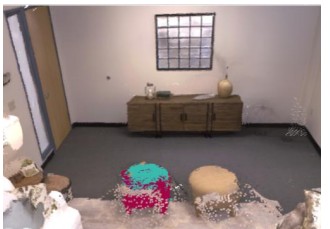 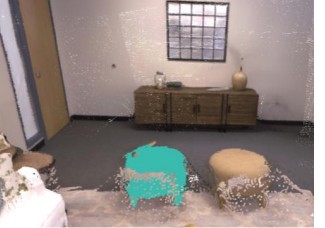

**Scene** **w/o IoU** **with IoU**

Figure 5: **Qualitative ablation study of the IoU metric.** Due to the sparsity of the pose graph, the same object may be incorrectly split into separate fragments over time when the IoU-based mechanism is disabled. The association mechanism based on the IoU metric can globally merge these fragments back into a single, consistent instance.

