# OpenReview forum: "Online 3D Instance Segmentation at task-oriented granularity with Unposed Monocular Video"
_ICLR.cc/2026/Conference — Submitted to ICLR 2026_

### Official Review · Reviewer_zBVZ · 2025-10-26

**Soundness:** 3
**Presentation:** 3
**Contribution:** 3
**Rating:** 4
**Confidence:** 5

**Summary:**

This paper presents a real-time framework for task-oriented 3D instance segmentation from unposed monocular video. Departing from traditional bottom-up segmentation, the method first detects task-relevant objects in 2D using an open-vocabulary detector and a prompt-based segmenter. These 2D masks are then efficiently fused into 3D instances by leveraging the pose graph and dense point correspondences from a modern monocular SLAM system (MASt3R-SLAM). A novel greedy clustering algorithm incorporates semantic and geometric cues for online multi-view mask association. The approach effectively mitigates the performance drop typically seen when using SLAM reconstructions instead of sensor depth, achieving competitive results on benchmarks and enabling open-vocabulary 3D segmentation in real-time.

**Strengths:**

1. The proposed method only requires unposed RGB stream as input, which is a practical setting.

2. The experiments are clear and sufficient.

**Weaknesses:**

1. The major concern of this paper is about efficiency. Inference latency should be reported for main experiments, which is essential for real-world SLAM and perception application.

2. Although this method can directly take in unposed RGB stream, it is more like a simple concatenation with Mast3R-SLAM and an online perception pipeline. The perception part cannot affect the SLAM process. It is expected that the semantic and instance information can help to refine the SLAM results.

**Questions:**

The inference speed should be reported in the experiments. Based on the rebuttal, I will consider to raise my score.

---

> ### Author Response · Authors · 2025-11-24
> **Official Comment by Authors**
>
> We thank the reviewer for their constructive comments. We have revised our manuscript according to the suggestions, and the detailed response is as follows:
> # W1&Q1: Regarding Running Efficiency
> Thank you for pointing this out. In the original submission, we reported the efficiency analysis in the appendix. In the revised version, we have moved the **Runtime Analyses section** into the main paper(see line411-421), as detailed below.
>
> We conduct runtime evaluation on a desktop computer equipped with an Intel i9-12900KS CPU and an NVIDIA RTX 3090 GPU. Our algorithm achieves 10.87 FPS on the scene office0 of the Replica dataset and 7.32 FPS on the scene0011_00 of the ScanNet dataset. For reference, Mast3r-SLAM runs at 11.23 FPS on Replica office0 and 7.58 FPS on ScanNet scene0011_00. Since our mask association module largely reuses the corresponding computation results from Mast3r-SLAM, both the association step and the system integration incur almost no extra computational overhead. Taking the scene office0 of the Replica dataset as an example, we report the runtime breakdown of all major components, as shown in the Tab.4. Specifically, the average processing time per keyframe is 30.1 ms for YOLO-World and 132.4 ms for SAM2, while the geometric association metric requires only 9 ms per keyframe pair, and the mask merging step takes 32 ms.
>
> # W2: Regarding the Integration of perception pipeline and SLAM
> Thank you for this insightful comment. Our response is as follows:
> Our work targets online, real-time, task-oriented 3D instance segmentation with only unposed monocular video as input. To meet robustness and latency requirements under open-vocabulary prompts, we currently adopt a perception–geometry weakly coupled architecture: a modern dense SLAM provides stable geometry and pose estimation, while the perception module adaptively controls instance granularity based on task prompts and efficiently performs cross-view mask association by reusing SLAM’s dense correspondences.
>
> Importantly, our system is not a simple concatenation. We leverage SLAM’s dense correspondences and pose graph to drive online multi-view mask association. Since these correspondences are constructed and maintained independently of global optimization variables (camera poses and the map), they are long-term valid and support incremental updates. Moreover, the sparsity of the pose graph reflects spatiotemporal correlations among keyframes, which avoids redundant frame-by-frame mask matching. As a result, our method achieves efficient and robust mask clustering with minimal overhead.
>
> We also agree that semantic/instance information can potentially benefit SLAM, and we view this as a promising extension. Possible directions include (but are not limited to): semantics-guided dynamic suppression, semantic loop-closure proposal generation, and object-level local reconstruction optimization. We will add a **Limitations and Future Works**section(Line519-527) in the revised version to discuss these directions in more detail, with the goal of inspiring subsequent research.

---

> > ### Comment · Reviewer_zBVZ · 2025-11-27
> >
> > Thanks for the authors' rebuttal. My concerns are solved and I will raise my score.

---

> > > ### Author Response · Authors · 2025-11-27
> > > **Acknowledgment to Reviewer zBVZ**
> > >
> > > We sincerely thank the reviewer for the constructive feedback and for raising the rating. We are glad that our responses were helpful in addressing the concerns.

---

### Official Review · Reviewer_UfRo · 2025-10-31

**Soundness:** 3
**Presentation:** 2
**Contribution:** 2
**Rating:** 4
**Confidence:** 3

**Summary:**

This paper proposes an online, task-oriented 3D instance segmentation framework for pose-free monocular videos. It leverages MASt3R-SLAM for dense reconstruction and pixel-level correspondence, while the front-end uses YOLO-World + SAM2 to generate task-relevant instance masks in 2D. And greedy clustering, supplemented by cluster-level 3D BBox-IoU merging. The authors report open-vocabulary/class-agnostic results on ScanNet200 and Replica, and analyze the performance degradation of existing methods when using SLAM reconstruction input.

**Strengths:**

1.Online scenarios and task-oriented settings have application value.

2.The results and failure reasons of ScanNet200 and Replica are analyzed and discussed.

**Weaknesses:**

1.Some hyperparameters were fixed at ϵ=0.2, GAM=0.25, SSM=0.85, and inter-cluster IoU=0.1, but no discussion or experimental analysis was performed.

2.Table 1 and Table 2 do not have any FPS comparison with the baseline, which cannot prove that the method is faster with the same accuracy, faster but with acceptable accuracy, or faster and accurate.

3.Table 1 and Table 2 do not compare Pose & Depth for GT.

**Questions:**

1.Section3.2  “given a task-relevant open-category set C” and “non-overlapping with small-mask retention” how this part is handled is not introduced in the paper.

2.The paper’s overall pipeline—2D mask cross-frame aggregation → 3D instance construction → text-conditioned selection is highly consistent with the core idea in Open3DIS, while its online, incremental cross-view mask integration overlaps with the OnlineAnySeg paradigm. The only substantive architectural deviation seems to be that mask association is restricted to SLAM pose-graph edges and that a greedy 3D IoU merge is used merely as a fallback. Could you analyze in more detail how this edge-restricted association differs from the multi-view matching strategy in Open3DIS and from the streaming merging in OnlineAnySeg, and in which scenarios this design may actually help?

3.The main architectural deviation seems to be that association is restricted to SLAM pose-graph edges and that a greedy 3D IoU merge is used as a post-hoc repair step. Could you clarify in more detail:

(a) how this SLAM-edge-restricted association differs, in practice, from the multi-view matching in Open3DIS and the streaming mask integration in OnlineAnySeg;

(b) whether this restriction could cause missed matches for long-range or sparsely observed objects; and

(c) whether the greedy 3D IoU backup can consistently recover such missed associations, or if there are scenarios where it fails?

[1] P. Nguyen, T. D. Ngo, E. Kalogerakis, C. Gan, A. Tran, C. Pham, and K. Nguyen, “Open3DIS: Open-vocabulary 3D instance segmentation with 2D mask guidance,” in Proc. IEEE/CVF Conf. Comput. Vis. Pattern Recognit. (CVPR), 2024, pp. 4018–4028.

[2] Y. Tang, J. Zhang, Y. Lan, Y. Guo, D. Dong, C. Zhu, and K. Xu, “OnlineAnySeg: Online zero-shot 3D segmentation by visual foundation model guided 2D mask merging,” in Proc. IEEE/CVF Conf. Comput. Vis. Pattern Recognit. (CVPR), 2025, pp. 3676–3685.

---

> ### Author Response · Authors · 2025-11-24
> **Official Comment by Authors**
>
> We greatly appreciate your careful review and constructive suggestions. Your comments have helped us improve the clarity of the paper. We have revised the manuscript accordingly, and our detailed responses are provided below.
>
> # W1： Regarding the Setting of Hyperparameters.
>
> Thank you for this feedback. We determined these hyperparameters through ablation studies and have added the corresponding analysis in the appendix A.3(**line 689-696**). The results show that our current settings are optimal and that the system’s performance is not sensitive to their exact values. The overall metrics vary only slightly across different settings, indicating the robustness of our method.
>
> # W2： Regarding the FPS comparison with the baseline.
> Thank you for this suggestion. In the revised version, we report FPS for all online baselines in Tables 1 (SAM3D: ~8 FPS, EmbodiedSAM: ~10 FPS, OnlineAnySeg: ~15 FPS)(**Table1**). Our system runs at ~7 FPS, which is slightly lower than these methods. However, note that our runtime is dominated by the SLAM module that estimates camera poses and the geometric map, whereas the compared methods use ground-truth poses and sensor point clouds and do not account for this cost.
>
> Our frame rate closely matches that of MASt3R‑SLAM (7.32 FPS vs. 7.58 FPS), indicating that our association step and system integration introduce almost no additional computational overhead. Please see the Runtime Analyses section for details.
>
> # W3: Table 1 and Table 2 do not compare Pose & Depth for GT.
> Thank you for the comment. Our method is inherently formulated for online unposed monocular video, unlike baselines that take ground-truth poses and depth as inputs, so we can’t report results under the GT Pose & Depth setting. However, we did evaluate baselines using SLAM‑estimated poses and depth instead of ground truth; under this setting, our method substantially outperforms these methods.
>
> # Q1: Regarding the explanation of “given a task-relevant open-category set C” and “non-overlapping with small-mask retention”:
> Thank you for pointing this out. We have added detailed explanations of both operations in the revised version.
>
> “The task-relevant open-category set C”(**line214-215**) refers to the set of semantic categories relevant to the task (e.g., {towel, bathtub} for “Find the towel on the bathtub.”). This set can be inferred/obtained automatically by an LLM.
>
> “non-overlapping with small-mask retention”(**Appendix:line671-682**): SAM2 may produce multiple instance masks per frame that partially overlap. Since each pixel should belong to exactly one instance, we compose a non-overlapping mask set Mi ∈ Z^{H×W} using a small-mask retention rule that preserves fine-grained instances (e.g., the towel will not be overwritten by the bathtub). We have incorporated this description into the main body of the revised version.
> Concretely, we sort all instance masks of the keyframe in descending area order (large to small). Then we write the sorted instance masks of a keyframe into a non-overlapping mask following this rule:
> (1) if the two masks share the same category label, we treat the smaller one as an over-segmentation part of the larger one and discard the the smaller mask;
> (2) if the two masks have different category labels, we keep the smaller mask on the overlapping pixels and truncate the larger mask accordingly.

---

> > ### Author Response · Authors · 2025-11-24
> > **Official Comment by Authors (to Q2 & Q3)**
> >
> > # Q2: Regarding the Differences between Our work and Prior methods in terms of the Text-conditioned segmentation paradigm, Multi-view matching strategy, and Streaming merging.
> > Thank you for this constructive question, which helps us clarify and emphasize our contributions. We have added this discussion to the Related Work sections(**line175-179**) in the revised manuscript.
> >
> > **Difference in our segmentation paradigm vs. Open3DIS**. Open3DIS first applies a superpoint segmentation on the point cloud, then projects superpoints back to 2D and uses their overlap with text-conditioned SAM masks to guide superpoint merging; the merged superpoints are taken as local 3D instances. In contrast, we directly use text-conditioned 2D instance masks and perform cross-frame instance association, without bottom-up superpoint merging—a more top-down segmentation paradigm.
> >
> > **Difference in our multi-view matching strategy vs. Open3DIS**. Open3DIS is a fully offline method that relies on fixed ground-truth point clouds and poses. Our setting is online monocular and unposed: camera poses and the map produced by dense SLAM are continuously refined by the backend. Multi-view matching that depends on these global variables is difficult to compute incrementally. Instead, we associate masks along SLAM pose-graph edges using dense pixel correspondences that are computed independently of global pose/map optimization and remain valid under back-end updates. This enables incremental association building and efficient online segmentation.
> >
> > **Difference in our streaming merging vs. OnlineAnySeg**. OnlineAnySeg performs online point‑cloud segmentation using multiple measures: spatial overlap and geometric similarity assume fixed ground‑truth point clouds, and third‑view consistency relies on accurate camera poses. In practice, with dense SLAM, both the map and poses are continuously updated by the back‑end, and OnlineAnySeg does not include corrections for these evolving optimization variables. Moreover, monocular pose/point‑cloud estimates inevitably contain noise. Even when using offline‑saved MASt3R‑SLAM poses and point clouds (thus ignoring dynamic updates), OnlineAnySeg remains vulnerable to such noise (see Table 1). As Reviewer RHEK noted, we instead use pose‑agnostic SLAM correspondences for association, which are unaffected by changes in poses or map variables, remain valid under back‑end updates, and can be built incrementally—making them more suitable for online use and more robust to noise in estimated poses and reconstructed geometry.
> >
> > # Q3 Regarding the (a) the differences between our SLAM-edge-restricted association and Open3DIS as well as OnlineAnySeg; (2) whether this restriction could cause missed matches for long-range or sparsely observed objects;(3) whether the greedy 3D IoU backup can consistently recover such missed associations, or if there are scenarios where it fails?
> >
> > Thank you for the thoughtful questions. Here are our responses to these three questions.
> >
> > (a) This question is identical to Question 2. We kindly ask the reviewer to refer to our previous response of Question 2.
> >
> > (b) Yes, SLAM-edge-restricted association may prune distant and weakly connected view pairs, which can avoid redundant, irrelevant inter‑frame matching. But this trade‑off may lead to miss matches for long‑range or sparsely observed objects.
> >
> > (c) Our greedy 3D IoU backup is designed to recover a portion of such missed links by merging 3D instances with high volumetric overlap after correspondence-driven grouping. However, there are scenarios where 3D IoU backup may fail:（1）Small, thin, or planar objects (e.g., long, slender items or thin surfaces) whose 3D bounding volumes are tiny or poorly estimated may yield low IoU even when they correspond to the same object; (2) Fragmented or partially reconstructed objects, where incomplete geometry underestimates overlap.

---

> > > ### Author Response · Authors · 2025-12-01
> > > **Supplementary for Q3**
> > >
> > > We provide a supplementary visualization analysis of the 3D IoU mechanism; please refer to new added Figure 5 in Appendix A.7.

---

> ### Author Response · Authors · 2025-11-28
> **Kind Follow-Up on Clarifications and Revisions**
>
> Thank you for your careful review and helpful feedback. We hope that the clarifications provided above, together with the additional experiments included in the revised manuscript, have adequately addressed your concerns. If you are satisfied, we kindly request you to consider updating your score or leaving a comment to reflect the newly added results and discussion. We remain committed to addressing any remaining points you may have during the discussion phase.

---

### Official Review · Reviewer_PNhw · 2025-11-01

**Soundness:** 3
**Presentation:** 2
**Contribution:** 3
**Rating:** 6
**Confidence:** 4

**Summary:**

This paper introduces a real-time, task-oriented 3D instance segmentation framework for unposed monocular video, designed to help embodied agents perceive and interact with objects in open-world environments. Unlike conventional bottom-up pipelines that segment before recognition, the method performs task-adaptive segmentation by combining an open-vocabulary detector with a prompt-based 2D segmenter, while reconstructing the scene’s 3D geometry online through a dense SLAM system. Leveraging the SLAM-derived pose graph, it efficiently associates and merges multi-view masks using geometric, semantic, and exclusivity cues through a greedy clustering algorithm. Experiments on open-vocabulary 3D instance segmentation benchmarks demonstrate strong performance, particularly on the Replica dataset, where the approach using only unposed images rivals methods that rely on ground-truth depth and poses.

**Strengths:**

1.This paper integrates a dense SLAM system with an online open-vocabulary instance segmentation framework, enabling 3D instance segmentation from only image sequences without requiring camera poses.

2. The method fuses and disentangles instance features across multiple frames, producing more stable and discriminative instance representations that lead to improved segmentation performance.

3.Extensive experiments on multiple datasets demonstrate the effectiveness of the proposed framework.

**Weaknesses:**

1.Figure 2 is difficult to interpret. The example only shows a single instance mask, and the components Mask Association Criteria and Online Clustering merely display matrices and instances without clearly illustrating the underlying mechanisms. Moreover, the depiction of the Pose Graph is overly abstract and does not intuitively convey its relation to camera poses. The final stage of the figure is also unclear—how the framework simultaneously produces both instance and semantic outputs remains unexplained.

2.Figure 2 claims that the method can output semantic segmentation results, yet the experiments do not include evaluations on standard semantic segmentation benchmarks.

3.In Equation (1), the “valid constraint”  V is not sufficiently explained. It would be helpful to clarify how the valid pixel correspondences are determined or obtained.

4.In Table 1, the comparison lacks fairness and completeness. The method SAM3D has an online version, and several other online approaches—such as EmbodiedSAM—are not included in the comparison.

5.Line 295 (page 6) states that the access and update operations on the matrix have an algorithmic complexity of O(1). A more detailed analysis or justification of this claim would make the explanation more convincing.

**Questions:**

1.In the ablation study, the MEM module appears to have the most significant impact. The paper mentions that “without MEM, inevitable point‑matching noise in the SLAM system” leads to performance degradation. Since the SLAM component is based on an existing network such as MASt3R‑SLAM, it would be valuable to clarify whether this issue originates from limitations inherent to the SLAM method itself or from the integration with the proposed framework.

2.It would be helpful to discuss the modularity of the proposed system. Specifically, can each component be replaced with functionally equivalent models—such as substituting VGGT‑SLAM for the SLAM backbone or using YOLO‑UniOW instead of YOLO‑World—without substantial adaptation?

3.The method relies on 2D models such as YOLO‑World and SAM, without additional training on 3D data or geometric structures. It would therefore be useful to elaborate on whether the approach demonstrates strong zero‑shot generalization capabilities in 3D perception tasks.

---

> ### Author Response · Authors · 2025-11-24
> **Official Comment by Authors**
>
> We sincerely appreciate the reviewer’s thoughtful and valuable feedback. We have revised the paper according to your comments, and our detailed responses are as follows.
> # W1： Figure 2 is difficult to interpret.
> Thank you for the helpful feedback. We have revised Figure 2, its caption, and related explanations in main body of the paper accordingly:
>
> (1)	We replaced the single instance mask with multiple instance masks.
>
> (2)	For the “Mask Association Criteria”, we added a detailed explanation in the text (**line308-311**): “These pairwise mask metrics are stored in a dense array, where the entry at row i and column j encodes the association score between the i‑th and j‑th masks.”
>
> (3)	We redesigned the pose‑graph subfigure to clarify its role. We primarily use inter‑frame correspondences indicated by pose‑graph edges. In the updated figure, circles denote keyframes and squares denote the correspondence blocks between keyframe pairs.
>
> (4)	For the final stage, we clarified in the caption(**line231-234**): “Followed by online clustering that establishes cross‑frame instance identities, the cross‑frame instance masks as well as corresponding semantic labels are back‑projected Into the reconstructed point cloud to obtain 3D instance and semantic maps.” We also added the text “back‑projected” inside the final arrow to make this step explicit.
>
> # W2: Regarding the evaluation of semantic segmentation.
> Thank you for this comment. To clarify, the “semantic segmentation map” shown in Figure 2 reflects per-instance semantic labels (i.e., each 3D instance is assigned a category). Our evaluations in Table 1 and Table 2 include open-vocabulary 3D instance segmentation metrics (AP@25/50), which inherently evaluate both instance segmentation and whether each instance’s category is correctly classified. In other words, the semantic correctness of instance labels is already accounted for in the reported metrics. We have added clarifications on the evaluation metrics in the revised version to avoid potential misunderstandings(**line 425-427**). Thank you for your feedback.
>
> # W3: Regarding explanation of “valid constraint” .
> Thank you for pointing this out. We have revised the wording in the revised version to make this part clearer.
>
> Specifically, the “valid constraint” V means that we only count pixels where the valid mask satisfies V_ij=True(**line270-271**). As defined above(**line194-196**), V_ij=False indicates invalid matches, such as those with large distances in 3D space or low predicted confidence.
>
> # W4: The comparison with Online SAM3D and EmbodiedSAM.
> Thank you for the suggestion. We have updated **Table 1** to include the online variant of SAM3D as well as EmbodiedSAM.
>
> We note that EmbodiedSAM is not a zero-shot method: it relies on semantic annotations from ScanNet200 for pretraining. Benefiting from extensive supervision aligned with the segmentation granularity of ScanNet200, EmbodiedSAM achieves substantially higher scores on the class-agnostic 3D instance segmentation evaluation compared to other zero-shot methods.
>
> # W5: Algorithmic complexity of accessing and updating operations on the matrix.
> Thank you for the helpful suggestion. We have reorganized and clarified this statement in the revised manuscript as follows:
>
> These pairwise mask metrics are stored in a dense array, where the entry at row i and column j encodes the association score between the i-th and j-th masks. Since random access to a dense array has O(1) time complexity, both lookups and updates of these entries can be performed in constant time, enabling efficient access and updates during online processing(**line307-311**).

---

> ### Author Response · Authors · 2025-11-24
> **Official Comment by Authors（Q1&Q2&Q3）**
>
> # Q1: What problems does MEM address, and how are these problems caused?
> Thank you for this helpful question. We have revised the wording in the updated manuscript to clarify this clearly.
>
> As stated in the paper, MEM primarily addresses the issue of “multiple distinct instances being erroneously merged into a single instance.” This issue is driven by two independent, module-intrinsic factors: (i) inevitable point-matching noise in the SLAM system, and (ii) a few under-segmented masks from the 2D segmentation module. These factors are independent, and either one alone can cause distinct instances to be erroneously merged(**line488-490**).
>
> # Q2：Discussion on different perception or SLAM methods
> Thank you for the suggestion. We have added this discussion in the appendix(**Appendix A.4**), as detailed below:
> Replacing the open-vocabulary detector. We directly replaced YOLO-World-xl with YOLO-UniOW-l (the largest publicly available variants of each family) without modifying any parameters of our system. On the Replica dataset, YOLO-World-xl is intrinsically stronger than YOLO-UniOW-l, so this substitution results in a moderate performance drop. Nevertheless, without any additional adaptation, the system remains fully functional and produces reasonable outputs, demonstrating that the detector module is readily replaceable.
>
> Replacing the SLAM backbone with VGGT‑SLAM. Like MASt3R‑SLAM, VGGT‑SLAM can perform online pose estimation and pointmap reconstruction from unposed monocular video. Concretely, VGGT‑SLAM partitions long sequences into overlapping submaps and performs direct submap registration/alignment; in its standard form, it does not expose frame‑to‑frame dense pixel correspondences, which our association module relies on. Therefore, a drop‑in replacement with VGGT‑SLAM is not currently supported. In principle, however, any SLAM that provides edge‑local dense correspondences (e.g., optical flow, feature matches, or pointmap matches) can serve as our backbone.
>
> # Q3： Statement on zero-shot setting
> Thank you for the suggestion. Our method indeed operates in a zero-shot setting and does not rely on training on 3D data or geometric structures. We have added clarification in the revised manuscript（**line102-103**） and labeled whether each baseline is zero-shot in **Tables 1** and **2**.
>
> While Replica is less challenging than ScanNet200 (cleaner data and fewer evaluated categories), we observe that several non–zero-shot methods (e.g., Open3DIS, OpenMask3D, Open‑YOLO 3D) transfer from their training dataset (ScanNet) to Replica dataset with nearly unchanged or only slightly decreased performance. In contrast, our method shows a substantial improvement on Replica, indicating strong zero-shot generalization capabilities in 3D perception tasks.

---

> > ### Author Response · Authors · 2025-12-01
> > **Supplementary for Q1**
> >
> > We provide a supplementary visualization analysis of the MEM mechanism; please refer to new added Figure 4 in Appendix A.7.

---

> ### Author Response · Authors · 2025-11-28
> **Kind Follow-Up on Responsesand Revisions**
>
> As the discussion phase is coming to a close, we kindly ask the reviewer to let us know whether the above responses have addressed the remaining questions. During the remainder of the discussion phase, we remain happy to respond to any additional points the reviewer may raise.
> We sincerely appreciate the reviewer’s effort and engagement throughout the process.

---

### Official Review · Reviewer_RHEK · 2025-11-01

**Soundness:** 2
**Presentation:** 2
**Contribution:** 2
**Rating:** 4
**Confidence:** 4

**Summary:**

This paper introduces a real-time, task-oriented 3D instance segmentation system for unposed monocular videos. The methodology departs from conventional bottom-up segmentation, instead coupling an open-vocabulary 2D segmentation (YOLO-World + SAM2) with a dense SLAM system (MASt3R-SLAM) for online 3D reconstruction. The core contribution is an online, greedy clustering algorithm that reuses the SLAM's internal dense correspondences (via GAM) and combines them with semantic similarity (SSM) and mutual exclusivity (MEM) metrics for multi-view mask merging.Evaluations on the ScanNet200 and Replica datasets show that the method is more robust to SLAM-induced artifacts compared to existing baselines, and maintains competitive performance without ground-truth poses or depth.

**Strengths:**

1. **Overall Strength**: This paper introduces a real-time, task-oriented 3D instance segmentation system for unposed monocular videos. The methodology departs from conventional bottom-up segmentation, instead coupling an open-vocabulary 2D segmentation (YOLO-World + SAM2) with a dense SLAM system (MASt3R-SLAM) for online 3D reconstruction. The core contribution is an online, greedy clustering algorithm that reuses the SLAM's internal dense correspondences (via GAM) and combines them with semantic similarity (SSM) and mutual exclusivity (MEM) metrics for multi-view mask merging. Evaluations on the ScanNet200 and Replica datasets show that the method is more robust to SLAM-induced artifacts compared to existing baselines, and maintains competitive performance without ground-truth poses or depth.
2. **Novel and clever Methodological Design**: The core technical idea is novel and clever. Reusing the internal, pose-independent dense correspondences from the SLAM system for mask association—rather than relying on the final, noisy 3D point cloud—is a sound strategy to bypass output-level noise.
3. **Well-Motivated Components**: The system's design is well-supported by empirical evidence. The ablation study (Table 3) provides clear justification for the three-metric association (GAM, SSM, MEM) and demonstrates that each component is necessary for the final performance.

**Weaknesses:**

1. **Fragile Pipeline and Critical Dependency on Upstream Modules**: The method entirely rely on MASt3R-SLAM for 3D reconstruction, and the segmentation performance heavily depend on the perfect functioning of its upstream components (MASt3R-SLAM, YOLO-World and SAM2). The paper fails to analyze how the system performs when the SLAM itself tracking is lost or provides highly noisy correspondences (e.g., in low-texture areas or during fast motion).The paper The paper fails to analyze the system's performance when either of these upstream modules fails or provides noisy input (e.g., SLAM tracking loss, low-texture scenes, 2D detection failures).
2. **Low Absolute Performance**: While the method is relatively more robust than failing baselines, its absolute performance (e.g., 6.5 AP50 on ScanNet200) is extremely low. This indicates that the proposed solution is only marginally functional and far from practical.
3. **Lack of Error Propagation Analysis**: The method is greedy and online. The paper does not discuss or analyze error accumulation. If an incorrect merge is made in an early frame, it's unclear if this error permanently poisons the instance cluster or if the system has any mechanism to recover.

**Questions:**

1. **Error Propagation**: Can your system recover from an incorrect merge made in an early frame (e.g., due to a 2D segmentation error), or do these errors propagate permanently?
2. **Failure Analysis**: The absolute performance is relatively low. Can you provide a more detailed failure analysis? For instance, what is the AP for small vs. large objects, or for head vs. tail classes? This would help clarify if the low score is due to SLAM/segmentation noise (as hypothesized) or systemic merging failures.
3. **Missing Demo Video**: Appendix A.2 (Reproducibility Statement) mentions a demo video available at an anonymous URL. Upon visiting this URL, I was only able to find a Readme file, and the demo video appears to be missing. Could the authors please verify the link and make the video available for review?
4. **Missing LLM Usage Statement**: ICLR submissions require a statement regarding the use of Large Language Models (LLMs). This paper appears to be missing this statement. Could the authors please clarify if LLMs were used in any part of the research or writing process?
5. **Minor Typo**: A minor point on presentation: In Equation (3), the formatting of the second norm $||s_m^j||$ appears to be malformed in the text.

I am open to raising my score if my concerns can be well addressed in the authors' rebuttal.

---

> ### Author Response · Authors · 2025-11-24
> **Official Comment by Authors**
>
> We sincerely thank you for carefully reviewing our paper and for providing many thoughtful comments. Your summary of the strengths has helped us clearly articulate the contributions of our work. Based on your feedback, we have made initial revisions to the manuscript and prepared a preliminary response. During the rebuttal period, we will continue to add experiments and refine our responses according to your suggestions.
>
> # W1: Fragile Pipeline and Critical Dependency on Upstream Modules
> Thank you for the insightful comment. We agree that our approach is zero-shot (no post-training on 3D data) and therefore benefits from advances in foundation models, while its performance is inevitably depended on the quality of upstream modules (SLAM, OV detection, 2D segmentation). We clarify robustness measures already in the system and will add further analyses.
>
> **Multi-criteria clustering**: We jointly use three criteria—GAM(Geometric association metric), SSM(Semantic similarity metric), and MEM (Mutual exclusivity metric)—for clustering. This multi-criteria design reduces reliance on any single noisy signal and limits erroneous merges.
>
> **SLAM tracking and correspondences**: When tracking fails, the system pauses cross-frame merging (no new edges/correspondences); upon relocalization, clustering is re-run with all available metrics. For correspondence quality, we filter correspondences using validity masks (low reprojection error and high confidence), ensuring that only reliable matches are used.
>
> **Detector/segmenter consistency checks**: We filter detections based on the consistency between YOLO-World boxes and SAM2 masks (low-overlap pairs are discarded), and apply intra-frame exclusivity to suppress under-segmentation conflicts.
>
> In short, while upstream quality influences the overall performance, these design choices—multi-criteria clustering (GAM/SSM/MEM), correspondence validation and SLAM relocalization, and detection–segmentation consistency checks—already mitigate single-module failures. We will further include remaining failure cases of the current approach in the appendix and provide corresponding analyses.
>
> # W2 Regarding Low Absolute Performance on ScanNet:
> Thank you for the feedback. We respond from two perspectives:
>
> **Problem difficulty and online constraints**. Our method tackles a significantly more challenging task: With unposed monocular video as input, we must first estimate camera poses and reconstruct 3D geometry online. Errors in the SLAM-based reconstruction inevitably affect the evaluation result. To meet real-time constraints, we reuse SLAM correspondences and add only minimal computation to achieve 3D instance segmentation online. This consideration of efficiency imposes additional accuracy challenges. Even when prior methods largely fail under the same SLAM result inputs, our system still produce reasonable output, demonstrating a viable path for online 3D instance segmentation from unposed monocular video.
>
> **Task-oriented objective vs. generic benchmarks**. Our focus is task-oriented 3D instance segmentation. While our AP on open-vocabulary benchmarks is lower than baselines that leverage GT depth and poses, our top-down segmentation paradigm better preserves task-relevant granularity and prevents over-merging of small objects (**see Fig.3**). In instruction-driven applications, the correctness of task-relevant granularity matters more than the AP metric on common benchmark(**line479-481**). We have made these two points clearer in the revised version.

---

> ### Author Response · Authors · 2025-11-24
> **Official Comment by Authors(W3, Q1-5)**
>
> # W3&Q1 Regarding Error Propagation
> Thank you for pointing this out. We agree that our use of the term “greedy” was misleading. Our association is not “greedy and irrevocable.” In Algorithm 1, “greedy” only refers to the processing order: candidate mask pairs are traversed in descending geometric association (GAM), so high‑confidence pairs are considered earlier, while mutual‑exclusivity (MEM) acts as a robust constraint. This is different from the classical “take a local optimum and commit irreversibly” notion.
>
> Concretely, suppose we have two candidate associations A–B (high confidence, correct) and A–C (low confidence, incorrect), and additionally an exclusivity relation between B and C. Our priority‑ordered procedure first merges A–B into one cluster. Because MEM records that B and C are mutually exclusive, C will subsequently be prevented from merging into the {A,B} cluster, even if the A–C edge exists. Thus, MEM blocks erroneous consolidation induced by low‑confidence links while preserving the correct high‑confidence merge.
>
> Importantly, the computation of association metrics(GAM/SSM/MEM ) is incremental, but the masks merging will be performed again with all available association metrics. Because the association/merge step is fast (<32 ms per run), whenever new edges added and the association metrics computed, our system will perform masks merging again under the full set of GAM/SSM/MEM constraints. Consequently, an early mistaken merge may be corrected later and does not propagate permanently.
> We have replaced the term “**greedy**” with “**priority-ordered**” in the revised version to avoid ambiguity. Thank you again for the helpful suggestion.
>
> ## Q2 Failure Analysis
> Thank you for the suggestion. We added a brief failure analysis by class frequency on ScanNet200:
>
> **Head classes**:        AP50/AP25 = 0.093 / 0.245
>
> **Common classes**:  AP50/AP25 = 0.055 / 0.174
>
> **Tail classes**:           AP50/AP25 = 0.041 / 0.135
>
> **Average**:                AP50/AP25 = 0.065 / 0.187
>
> The gap between AP50 and AP25 is large across all groups, and the drop is most pronounced for common/tail classes. This pattern is consistent with our hypothesis: SLAM/reconstruction noise and occasional 2D under-/over-segmentation reduce 3D instance segmentation performance (especially AP50), with stronger impact on small/long-tail objects. We will report these in the Appendix(**A.6**).
>
> # Q3 Missing Demo Video
> Thank you for pointing this out. We will upload the demo video and fix the link as soon as possible.
>
> # Q4 Missing LLM Usage Statement
> Thank you for the reminder. We realized that we mistakenly selected “No, not at all” for LLM usage during paper registration. We did use an LLM for language polishing during manuscript preparation, and we have added the following statement in the appendix(**A.5.**):
>
> Large language model (LLM) were used as an auxiliary tool during the writing and language refinement of this manuscript. Its role was limited to improving linguistic expression, readability, and overall flow, including tasks such as sentence rephrasing and grammar checking.
>
> It is important to emphasize that the LLM did not participate in the research conception, methodological design, or experimental procedures. All research ideas, technical approaches, and data analyses were independently developed and conducted by the authors. The LLM’s contribution was strictly confined to language-level improvements and did not involve any scientific content.
>
> The authors take full responsibility for the entire manuscript, including the portions generated or refined with LLM assistance. We have ensured that all content complies with academic ethical standards and does not involve plagiarism or any form of scientific misconduct.
>
> # Q5 Minor Typo
> Thank you for the careful reading and for pointing out this typo. We have corrected the formatting issue in **Equation (3)**.

---

> > ### Author Response · Authors · 2025-11-27
> > **Response to Q3**
> >
> > ## Q3 Missing Demo Video
> > We have uploaded the demo video and fixed the link in the Appendix A.2. We sincerely invite the reviewers to download and watch the videos.

---

> ### Comment · Reviewer_RHEK · 2025-11-28
>
> Thanks for the detailed rebuttal. My concerns have been addressed, and I will raise my score as promised.

---

> > ### Author Response · Authors · 2025-11-28
> > **Acknowledgment to Reviewer RHEK**
> >
> > Thank you very much for your careful review and insightful feedback, and we also appreciate your participation during the rebuttal period. We are pleased that we were able to address your concerns.

---

### Author Response · Authors · 2025-11-28
**Official Comment by Authors**

Thank you to all reviewers for the careful review of our paper and for providing many constructive suggestions. During the rebuttal period, we incorporated additional experiments and revisions based on the reviewers’ feedback, and we believe this process has indeed improved the quality of our paper.

Reviewers zBVZ and RHEK indicated that our responses had addressed their concerns and expressed willingness to raise their scores. However, **due to the system closure, it appears that RHEK was unable to update the score in the system**. The other two reviewers (PNhw and UfRo) have not yet had the opportunity to provide further responses.

This year’s unexpected incident has caused significant disruption for everyone. We would like to emphasize that throughout this special period, we strictly followed all official ICLR guidelines during the rebuttal. We sincerely appreciate all reviewers for their feedback and participation, and we extend our heartfelt gratitude to the ACs and PCs for taking on additional responsibilities and coordination during this urgent situation.

Before the discussion period ends, we remain fully willing to address any new comments.

---

### Author Response · Authors · 2025-12-02
**A summary to PCs and ACs**

Dear PCs and ACs,

We would like to express our sincere gratitude to ICLR PCs for their timely response to this urgent incident, and to the ACs for dedicating significant time and effort to manage the review process.

We also thank the four reviewers for their careful evaluations and constructive feedback. Their comments indeed helped us improve the quality of our paper during the rebuttal stage. We provided point-by-point responses (see below) and submitted a revised version with changes highlighted in **blue**, and with each response specifying the corresponding modification locations in the revised paper (in **bold black**).

Given the heavy workload and tight schedule for ACs, we summarize below the key concerns of each reviewer, along with our rebuttal responses and interactions.

Our initial scores were 6, 4, 4, and 4. **Reviewer zBVZ (initial rating: 4 / confidence: 5) has raised his/her score to 6 during the rebuttal. Reviewer RHEK (initial rating: 4 / confidence: 4) stated in his/her response: *“My concerns have been addressed, and I will raise my score as promised.”***  Reviewers **UfRo** (initial rating: 4 / confidence: 3) and **PNhw** (initial rating: 6 / confidence: 4) had not yet replied during the rebuttal.

- Reviewer zBVZ’s main concern was system efficiency(W1). In fact, we had already included an efficiency analysis in the appendix of the initial submission, reporting our system’s FPS (7.32 FPS on ScanNet, 10.87 FPS on Replica) and the runtime of each core module. As requested, we moved this analysis into the main text and additionally reported that our system introduces almost no extra overhead compared to our SLAM backbone (Mast3r-SLAM): 7.32 FPS vs. 7.58 FPS on ScanNet, and 10.87 FPS vs. 11.23 FPS on Replica. We also clarified the relationship between our perception module and the SLAM module and added potential research directions in the Limitations and Future Work section(W2). **Reviewer zBVZ raised his/her score from 4 to 6.**



- Reviewer RHEK acknowledged that **our core technical idea is novel and clever (see Strengths2)** and **indicated willingness to raise his/her score if concerns were addressed in the initial review comments**. His/her concerns included:

  1. System robustness. We responded to each issue raised and explained the  corresponding mechanisms used in our system (see W1)；
  2. Relatively low ScanNet benchmark metrics. We explained that unlike prior methods, we adopt a setting that is more aligned with real-world applications but inherently more challenging. Under this harder setting, our method still clearly outperforms existing approaches. We also emphasized that beyond generic benchmark metrics, our focus is on achieving the correct segmentation granularity required by the task-oriented objective in practical scenarios (see W2 and our responses).
  3. Error accumulation concerns . In our initial submission, we described our method using the term “greedy mask clustering”, which misled the reviewers into believe that our method is “greedy and irrevocable.” We clarified that the actual mechanism of our method is not irrevocable, and does not incur error accumulation. To avoid misunderstanding, we replaced “**greedy**” with “**priority-ordered**” in the revised version (see W3 & Q1).

  We also provided detailed AP metrics(Q2), a demo video(Q3), and the LLM Usage Statement as requested(Q4). **Reviewer RHEK indicated:  *“My concerns have been addressed, and I will raise my score as promised.”***



- Reviewer UfRo’s main concerns were the lack of hyperparameter analysis (W1) and missing FPS reports for baseline methods (W2). We added ablations on hyperparameters in the appendix, showing that our current settings are generally optimal and that the system is not sensitive to these parameters. We also added FPS results for baselines in Table 1. Regarding the missing GT depth/pose setting (W3), we clarified that our method is designed for online unposed monocular video, so we cannot report GT depth/pose metrics for our own method, but we reported both settings for baselines. Reviewer UfRo’s Q2 touches on our key innovation (also highlighted by Reviewer RHEK), and we provided a detailed explanation (see Q2) and expanded the discussion in the related work section of revised version. We also addressed Q1 and Q3 and revised the corresponding content of paper accordingly, and additionally provided the corresponding visual analysis for Q3 in the appendix.



- Reviewer PNhw's main concerns were unclear description for some figures and descriptions. We revised all relevant parts of the main text as requested (W1, W3, W5). We clarified that the semantic evaluation is already included in the main text (W2) and added the requested baseline comparisons(W4). We responded to Q1 and Q3 with clarifications in the text, supplemented Q1 with visual analysis in the appendix, and added experiments and discussion for Q2 in the appendix.

---

> ### Author Response · Authors · 2025-12-02
> **A summary to PCs and ACs**
>
> We sincerely thank the ACs for their time and effort. Despite the unexpected incident during the rebuttal period, **we  solemnly affirm that we have strictly  and fully followed all official ICLR policies throughout the process**.  We have actively addressed and clarified all reviewers’ questions, and supplemented our responses with additional qualitative and quantitative experiments. We trust that the AC will make a fair judgment based on the available material, and we fully respect and support any decision made by the ACs and the ICLR PCs.

---

### Meta-Review · Area_Chair_uMQz · 2026-01-07

**Summary:**

The paper presents a system for online, real-time, referring expression based 3D instance segmentation from unposed monocular video. The core idea is to integrate MASt3R-SLAM with a pose graph representation and object masks produced by SAM, and to associate these object masks over time. The system demonstrates promising results on two public datasets.

The initial reviews were mixed. While reviewers found the key idea interesting and considered the experimental evaluation sufficient, they raised several concerns, including: (1) the robustness of the system to potential failures in individual modules; (2) lack of clarity in certain components of the method; (3) the risk of error accumulation over time; (4) comparisons with recent Open3DIS and OnlineAnySeg; and (5) justification of the reported performance levels.

After going through the paper, the reviews, and the rebuttal, the initial decision is to recommend rejection. This decision is primarily based on the outstanding concern regarding technical contribution. The authors are encouraged to incorporate review feedback, revise the paper, and resubmit to a different venue.

**Reviewer Concerns:**

Some of the concerns have been well addressed in the rebuttal. In particular, two reviewers (zBVZ and RHEK) indicated that they intend to raise their scores following the authors' clarifications and additional analysis.

However, a major remaining concern relates to the conceptual distinction from Open3DIS and OnlineAnySeg. While the overall setup (unposed RGB video vs. RGB-D input) and specific technical details differ, as noted by Reviewer UfRo, the mask association component appears conceptually similar to that of OnlineAnySeg, and the text-guided segmentation component is closely related to Open3DIS. As a result, it remains somewhat unclear whether these components introduce substantial new technical contributions beyond existing approaches.

Another minor issue is that the demo videos updated during the rebuttal (linked from the appendix) appear to be corrupted and cannot be viewed, which makes it difficult to fully assess the qualitative results.

**Reviewer Scores:**

Reviewer PNhw raised only minor comments and is likely to maintain their score. It is difficult to predict Reviewer UfRo’s final rating. While the rebuttal addressed most of UfRo’s concerns, the question regarding the relationship to Open3DIS and OnlineAnySeg remains partially unresolved.

---

### Decision · Program_Chairs · 2026-01-26

Reject